

# Identification of microRNA signatures in umbilical cord blood associated with maternal characteristics

Jaroslav Juracek[1], Pavel Piler[2], Petr Janku[3], Lenka Radova[1] and Ondrej Slaby[1]

[1] Central European Institute of Technology (CEITEC), Masaryk University, Brno, Czech Republic
[2] Research Centre for Toxic Compounds in the Environment (RECETOX), Faculty of Science, Masaryk University, Brno, Czech Republic
[3] Department of Gynecology and Obstetrics, Institutions shared with the Faculty Hospital Brno, Institutions of Reproductive Medicine, Faculty of Medicine, Masaryk University, Brno, Czech Republic

## ABSTRACT

**Background**. Umbilical cord blood could serve as useful source of blood markers enabling more efficient and reliable prenatal and neonatal diagnostics. MicroRNAs (miRNAs) are ubiquitous in body fluids where they were used for detecting and monitoring various physiological and pathological conditions. In this descriptive study, we aimed to identify changes in miRNA expression profiles associated with basic maternal somatic and epidemiological characteristics.

**Methods**. Study is based on 24 mothers from the Pilot phase of CELSPAC: TNG (Central European Longitudinal Studies of Parents and Children: The Next Generation) study. Cord blood was collected at time of delivery and global miRNA profiling was performed using microRNA Ready-to-use PCR Human Panel I+II TaqMan microarrays. Expression profiles were statistically evaluated in relation to maternal age, BMI, pregnancy weight gain, blood type, Rh factor status, allergies during pregnancy, addictive substance abuse and smoking status.

**Results**. We analyzed expression of 752 human mature miRNAs in 24 samples of umbilical cord blood. For all maternal characteristics tested we described a specific signature of significantly deregulated miRNAs ($P < 0.05$). Analysis revealed seven miRNA associated with maternal age (three increased and four decreased in women younger than 35 years), 14 miRNAs associated with BMI status (five miRNAs increased and nine miRNAs decreased in women with BMI $> 25$) and nine miRNAs associated with maternal weight gain during pregnancy (eight miRNAs increased, and one miRNA decreased in women with weight gain $< 12$ kg). Additionally, 17 miRNAs correlated to blood type (two miRNAs decreased in blood type A, 11 increased in blood type B, two miRNAs increased in blood type AB and two miRNAs increased in blood type 0) and 17 miRNAs to Rh status of mother. We also detected seven miRNAs deregulated in umbilical cord blood of women with allergy (four increased and three decreased in women with allergy), four miRNAs associated to addictive substance abuse status (two up- and two downregulated in women with addictive substance abuse) and eight miRNAs associated with maternal cigarette smoking during pregnancy.

**Conclusions**. We successfully described differences in miRNA profiles in umbilical cord blood associated with basic characteristics connected with mother. Our data suggest that miRNAs in umbilical cord blood are detectable and associated with a wide range of maternal characteristics. These results indicate that miRNAs could potentially serve, and

Corresponding author
Ondrej Slaby,
ondrej.slaby@ceitec.muni.cz,
on.slaby@gmail.com

should be studied, as biomarkers for screening and diagnosis of pregnancy-associated complications and pathologies.

## INTRODUCTION

Umbilical cord blood (UCB) is blood that remains in the placenta and umbilical cord after birth (*Waller-Wise, 2011*). Apart from common blood elements cord blood is a rich source of primitive, undifferentiated hematopoietic stem cells (*Erices, Conget & Minguell, 2000*). Though it was originally considered as a waste product, it has developed into an important allogeneic donor source in transplantation in pediatrics and a novel source of blood markers (*O'Brien, Tiedemann & Vowels, 2006*). Especially in neonatal diagnostics, where blood from peripheral veins is used, UCB might be a suitable alternative and valuable source of blood biomarkers thanks to noninvasive and painless collection. Recent studies show that certain acute phase reactants are elevated in umbilical cord blood of premature infants with early onset sepsis (*Mithal et al., 2017*). Other studies described the distribution of immune biomarkers in cord blood across gestational age and show the association between biomarker level patterns and preterm birth (*Matoba et al., 2009*). Similarly, growth factors levels in cord blood can correlate with birth weight and postnatal growth in premature infants and was also associated with risk for postnatal growth failure (*Voller et al., 2014*).

One of the most abundant groups of biomarkers are microRNAs (miRNAs). They are ubiquitous in most of the body fluid types, where they may have functional roles associated with the surrounding tissues (*Weber et al., 2010*). In addition, the changes in levels of specific miRNAs in body fluids were used for detecting and monitoring various somatic and pathological conditions (*Cortez et al., 2011*; *Velu, Ramesh & Srinivasan, 2012*). The role of circulating miRNAs has been reported also in the context of neonatal diagnostics. Higher expression levels of miR-615-3p were observed in neonatal peripheral blood where this miRNA promoted acute respiratory distress syndrome (ARDS) development (*Wu & Ding, 2018*). Similarly, a decrease in levels of miR-132 and miR-223 was associated with neonatal sepsis (*Dhas, Dirisala & Bhat, 2018*).

The first description of miRNA profiling in cord blood was reported in 2015. In this study, downregulation of miR-374a-5p was observed in infants with hypoxic ischemic encephalopathy (HIE) (*Looney et al., 2015*). In a subsequent study, altered miRNA levels were detected also in umbilical cord blood of neonates with perinatal asphyxia (PA), suggesting their potential role in early detection of this disease (*O'Sullivan et al., 2018*). Moreover, previous research has shown that not only pathologies but also somatic characteristic such as birth weight modifies the expression of miRNAs (*Rodil-Garcia et al., 2017*). Based on these observations, we hypothesize that specific miRNA patterns in umbilical cord blood can be associated with physiological as well as pathological conditions of mother or fetus.
_______________________________________

In this pilot study, we aimed to perform high-capacity screening of miRNA levels in umbilical cord blood in order to describe miRNA signatures associated with basic maternal somatic and epidemiological characteristics.

## MATERIALS & METHODS

### Patient cohorts

Study data and samples were obtained from the Pilot phase of CELSPAC: TNG (Central European Longitudinal Studies of Parents and Children: The Next Generation Study). CELSPAC: TNG is designed as a new prospective birth cohort which will follow up on 10,000 children from their prenatal period to adolescence with the aim of assessing multiple factors potentially affecting children's health. The Ethical committee of University Hospital Brno, Czech Republic approved this study (No. 20140409-01). All mothers gave their written informed consent.

The Pilot phase of CELSPAC: TNG was initiated in April 2015 to evaluate feasibility of the protocol for collection, processing and storing of biological samples (cord blood; venous blood, urine and buccal smear from mothers; stool, dry blood spot and buccal smear from babies); to estimate future study response rates; to evaluate on-line distribution and respond rate of questionnaires.

The current study included 24 mothers from whom we collected umbilical cord blood (UCB) at time of delivery. We also used the data from medical records related to mother somatic characteristics, pregnancy and birth. Women with non-physiological pregnancy including medical and obstetrical complications or comorbid conditions that could affect fetus were excluded.

### Sample collection and processing

Cord blood was collected to S-Monovette® K3E (S-Monovette® 9 ml, K3 EDTA) after the second stage of labor from the umbilical cord vein. Plasma was prepared by centrifugation (2,500 g for 10 min at 22 °C) and aliquoted into tubes as 250 µL samples and stored at −80 °C.

### miRNA Quantification

Prior to RNA isolation umbilical cord blood plasma samples were centrifuged at 4 °C at 1,000 g for 5 min. Total RNA from 200 µl of UCB plasma was isolated using miRNeasy Serum/Plasma Kit (Qiagen, Hilden, Germany). RNA quality and quantity was evaluated using Nanodrop 2000 Spectrophotometer (Thermo Fisher Scientific, Waltham, MA, USA). Whole-genome miRNA profiling was performed by use of Human panel I+II, V4, miRCURY LNA miRNA miRNome PCR Panel (Exiqon by Qiagen, Hilden, Germany) accordingly to the manufacturers protocol. On each plate for each sample we included interplate calibrator enabling compensation of signal variations between instrument runs (inter-plate calibrator assay UniSp3). Expression levels of miRNA represented by Ct values were normalized on U6 reference gene expression level using the $2^{-\Delta Ct}$ method where $\Delta Ct = (Ct\_Target\ miRNA − Ct\_U6)$. Relative miRNA expression levels were correlated with selected epidemiological and somatic characteristics of mothers. Only miRNAs having

**Table 1  List of maternal characteristics used in specific statistical analysis.**

| Maternal characteristics | Specific characteristics (Number of subjects) | | | |
|---|---|---|---|---|
| Age | <35 years (15) | | >35 years (9) | |
| BMI | <25 (19) | | >25 (5) | |
| Pregnancy weight gain | <12 kg (12) | | >12 kg (12) | |
| Blood type | A (10) | B (4) | 0 (2) | AB (5) |
| Rh factor | Negative (6) | | Positive (18) | |
| Allergies | Yes (12) | | No (12) | |
| Addictive substance abuse | Yes (9) | | No (15) | |
| Smoking status | Non-smoker (15) | | Smoker (3) | Stop-smoker (6) |

non-zero expression values within more than 50% of samples were included in statistical analysis. Statistical analyses were performed within R/Bioconductor environment. The Mann–Whitney and Kruskal–Wallis test were applied for two or more categorial variables. In all comparisons, $p$-values <0.05 were set as statistically significant.

# RESULTS

Global profiling performed using TaqMan array enabling detection of 752 human mature miRNAs was performed in 24 umbilical cord blood plasma samples (Dataset S1). From this number, 656 miRNAs were detectable in at least one sample and 491 miRNAs had non-zero expression values within more than 50% of samples. For subsequent statistical analysis, samples were regrouped by selected maternal characteristics including age, BMI, pregnancy weight gain (PWG), blood type, Rh factor status, allergies during pregnancy, addictive substance abuse and smoking status (summarized in Table 1). Genome-wide microRNA expression data have been deposited in the Gene Expression Omnibus repository under accession number GSE128943.

Global expression analysis revealed pattern of seven miRNA associated with maternal age namely miR-137, miR-665, miR-770-5p were increased and miR-625-3p, miR-377-3p, miR-224-3p and miR-671-3p decreased in women younger than 35 years. Next, we analyzed miRNA changes in relation to a maternal BMI. Analysis identified 14 miRNAs with differential levels between women with overweight (BMI > 25) and women with normal BMI status (BMI 18.5–25). Five of these miRNAs were increased and nine miRNAs decreased in women with overweight (BMI > 25). Similarly, levels of nine miRNAs were amended in umbilical cord blood in association with maternal weight gain during pregnancy (eight miRNAs increased, and one miRNA decreased in women with pregnancy weight gain lesser than 12 kg). Additionally, we identified 17 miRNAs with levels correlated to blood type of mother (two miRNAs showed decreased levels in blood type A group, 11 showed increased levels in blood type B group, two miRNAs were increased in blood type AB group and two miRNAs were increased in blood type 0 group). Moreover, 17 miRNAs were significantly increased when samples of Rh positive and Rh-negative women were compared. In mothers with unspecified allergies we detected pattern of seven miRNAs with deregulated levels; miR-181d-5p, miR-545-3p, miR-153-3p, miR-632 increased in women with allergy, miR-371a-3p, miR-96-5p, miR-216a-5p decreased in women with allergy. Next, we described four miRNAs with levels associated to addictive substance

**Table 2** List of umbilical cord blood miRNAs significantly associated with individual maternal characteristics ($P < 0.05$).

| Maternal characteristics | List of associated miRNAs |
|---|---|
| Age(*<35 years*) | ↑ miR-137, miR-665, miR-770-5p; ↓ miR-625-3p, miR-377-3p, miR-224-3p, miR-671-3p. |
| BMI(*>25*) | ↑ miR-1203, miR-143-3p, miR-582-5p, miR-510-5p, miR-450a-5p; ↓ miR-604, miR-205-5p, miR-551a, miR-203a, miR-548l, miR-424-5p, miR-627-5p, miR-629-3p, miR-141-3p. |
| Pregnancy weight gain(*<12 kg*) | ↑ miR-138-5p, miR-760, miR-9-3p, miR-548c-5p, miR-1260a, miR-145-3p, miR-34a-3p, miR-320d; ↓ miR-1224-3p. |
| Blood type | A: ↓ miR-380-5p, miR-92a-1-5p;B: ↑miR-760, miR-10b-5p, miR-34b-3p, miR-145-5p, miR-153-3p, miR-548c-5p, miR-511-5p, miR-330-5p, miR-24-1-5p, let-7b-3p, let-7f-2-3p; AB: ↑ miR-595, miR-431-3p; 0: ↑ miR-641, miR-548h-5p. |
| Rh factor status(*positive*) | ↑ miR-141-3p, miR-188-5p, miR-211-5p, miR-205-5p, miR-150-5p, miR-181c-5p, miR-124-3p, miR-142-5p, miR-15b-5p, miR-1269a, miR-1260a, miR-518d-3p, miR-27a-5p; ↓ miR-514a-3p, miR-449b-5p, miR-641, miR-548l. |
| Allergies(*Yes*) | ↑ miR-181d-5p, miR-545-3p, miR-153-3p, miR-632; ↓ miR-371a-3p, miR-96-5p, miR-216a-5p. |
| Addictive substance abuse(*Yes*) | ↑ miR-138-1-3p, miR-33b-3p; ↓ miR-760, miR-377-3p. |
| Smoking status(*Yes*) | ↑ miR-129-5p, miR-30b-3p, miR-187-3p, miR-507, miR-520b, miR-33b-3p, miR-138-1-3p; ↓ miR-760. |

abuse status. Subsequent multivariate statistical analysis revealed increased levels of six miRNAs (miR-129-5p, miR-30b-3p, miR-187-3p, miR-507, miR-520b and miR-33b-3p) associated with maternal cigarette smoking during pregnancy. Similarly, we identified significantly higher level of miR-138-1-3p and decreased level of miR-760 in mothers who quit smoking during pregnancy. Complete list of deregulated miRNAs associated with monitored maternal characteristics is summarized in Table 2 (detailed information is Tables S1–S8).

## DISCUSSION

Circulating miRNAs are currently widely accepted as promising markers of both pathological and physiological conditions (*Velu, Ramesh & Srinivasan, 2012*; *Kosaka, Iguchi & Ochiya, 2010*; *Mitchell et al., 2008*). In reproductive medicine specific miRNA profiles in peripheral blood were found to relate to complications of pregnancy, such as placental abruption (*Miura et al., 2017*), ectopic pregnancy (*Zhao et al., 2012*) or preeclampsia (*Gunel et al., 2011*). Moreover, expression patterns of circulating miRNA are promising solution for noninvasive prenatal testing of Down Syndrome and other genetic diseases (*Erturk et al., 2016*). However, considering direct connection with fetus, umbilical cord blood could serve as valuable source of biomarkers in prenatal diagnostics and screening.

In our pilot study, we successfully described aberration in miRNA profiles in umbilical cord blood plasma associated with basic characteristics connected with mother. So far publications focused on the identification of miRNA expression profiles in umbilical cord blood were connected mainly to pathophysiology of a particular disease or pathological

conditions. As in the case of infants with hypoxic ischemic encephalopathy (HIE) where miR-374a revealed significant down-regulation in cord blood of infants with perinatal asphyxia and subsequent HIE (*Looney et al., 2015*). Similarly study of Rager et al. highlight miRNAs as novel responders to prenatal arsenic exposure that may contribute to associated immune response perturbations (*Rager et al., 2014*). Despite diagnostic potential of UBC there is lack of descriptive studies which demonstrate miRNA deregulation in association with basic somatic and epidemiological characteristics of mothers and newborns. Ghaffari et al. investigated whether maternal obesity is associated with alterations in expression of fetal miRNAs (*Ghaffari et al., 2015*). Despite negative results, this study delineated role of miRNA within delivery course and success rate.

Since umbilical cord blood flow is dynamic and progressive process where exchange of blood elements and nutrients occurs, we expected that possible deregulation should be influenced by both the mother's and the newborn's environment. Currently there are no comparable studies supporting our findings, however, we found overlap in identified miRNAs within studies focusing on miRNAs functioning. For example, miR-625-3p and miR-671-3p showing significant association with maternal age in our study were described also in study of Huan et al. as age-associated miRNAs (*Huan et al., 2018*). Moreover, miR-671-3p seems to be differentially expressed between keratinocytes prepared from child skin and aged skin (*Muther et al., 2017*). Similarly, we identified miRNAs indicating association with maternal BMI or weight gain during pregnancy. In accordance with our findings, miR-143-3p was depicted as regulator of adipocyte differentiation (*Esau et al., 2004*) and its upregulation in mesenteric fat in mice was associated with body weight (*Takanabe et al., 2008*). MiR-450-5p, miR-203a, miR-141-3p and miR-205-5p were found to be differentially expressed in subcutaneous adipose tissue of obese individuals and normal-weight subjects (*Kurylowicz et al., 2018*). Other miRNAs such as miR-551a or miR-138-5p were already associated with BMI (*Iacomino et al., 2019*) or weight gain (*Zhao et al., 2017*). Regarding to mothers with allergies, we identified miR-181d-5p, member of miR-181 family, which has a central role in vascular inflammation by controlling critical signaling pathways and regulates immune cell homeostasis (*Sun, Sit & Feinberg, 2014*). Other identified molecule, miR-371a-3p was suggested to modulate the Th1/Th2 balance in asthma (*Qiu et al., 2017*). Among miRNAs deregulated within mothers who did smoke during pregnancy we described miR-129-5p which was observed to be upregulated in lung cancer patients with a smoking history (*Momi et al., 2014*; *Vucic et al., 2014*). Further, miR-33b-3p was differentially expressed between rectal cancer tissue and normal rectal mucosa and associated with smoking and miR-520b was significantly differentially expressed with cigarette smoking and associated with CIMP and/or MSI status in colon and rectal cancer (*Mullany et al., 2016*).

## CONCLUSION

The data obtained in this pilot study indicated that miRNA levels in umbilical cord blood plasma are related to somatic and epidemiological characteristics of mother and newborn infants. Therefore, UCB miRNAs should be studied also as biomarkers for screening and diagnosis of pregnancy-associated complications and pathologies.

## ACKNOWLEDGEMENTS

Many thanks go to the participating families as well as the physicians and medical staff of The University Hospital Brno, and the entire study team.

### Funding

This study was funded by the Ministry of Education, Youth and Sports of the Czech Republic and European Structural and Investment Funds (CETOCOEN PLUS project: CZ.02.1.01/0.0/0.0/15_003/0000469 and the RECETOX research infrastructure: LM2015051) and by the Ministry of Health, the Czech Republic (FNBr, 65269705). The funders had no role in study design, data collection and analysis, decision to publish, or preparation of the manuscript.

### Grant Disclosures

The following grant information was disclosed by the authors:
Ministry of Education, Youth and Sports of the Czech Republic.
European Structural and Investment Funds: CETOCOEN PLUS project: CZ.02.1.01/0.0/0.0/15_003/0000469.
RECETOX research infrastructure: LM2015051.
Ministry of Health, the Czech Republic: FNBr, 65269705.

### Competing Interests

The authors declare there are no competing interests.

### Author Contributions

- Jaroslav Juracek performed the experiments, analyzed the data, prepared figures and/or tables, authored or reviewed drafts of the paper, approved the final draft.
- Pavel Piler conceived and designed the experiments, performed the experiments, contributed reagents/materials/analysis tools, approved the final draft.
- Petr Janku performed the experiments, analyzed the data, contributed reagents/materials/analysis tools, approved the final draft.
- Lenka Radova conceived and designed the experiments, analyzed the data, prepared figures and/or tables, approved the final draft.
- Ondrej Slaby conceived and designed the experiments, analyzed the data, contributed reagents/materials/analysis tools, prepared figures and/or tables, authored or reviewed drafts of the paper, approved the final draft.

### Human Ethics

The following information was supplied relating to ethical approvals (i.e., approving body and any reference numbers):

The Ethical Committee of University Hospital Brno, Czech Republic approved this study (No. 20140409-01).

## Data Availability

The raw measurements are available in Dataset S1 and Table S9. In Table S9, all samples are stratified into the clinico-epidemiological categories tested in the study (e.g., age, pregnancy weight gain, blood type, Rh factor, etc.). In Dataset S1, the global expression profiles of microRNAs are presented as Ct values for each sample.

## Supplemental Information

Supplemental information for this article can be found online at http://dx.doi.org/10.7717/peerj.6981#supplemental-information.

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
