# Peer review of "Identification of microRNA signatures in umbilical cord blood associated with maternal characteristics"

_PeerJ, doi:10.7717/peerj.6981_

## Round 0.1 · original submission · Minor Revisions

The authors could discuss some studies that show the application of pregnancy-related microRNAs in diagnosis or screening.

Reviewer 1 ·

Basic reporting

None

Experimental design

None

Validity of the findings

None

Additional comments

The manuscript by Juracek et al., “Identification of microRNA signatures in umbilical cord blood associated with maternal characteristics” demonstrated the UCB miRNAs could serve as biomarkers for pregnancy-associated complications. The content of this study is of greater interest and suitable for publication in the PeerJ. I recommend the authors could discuss some studies that show the application of pregnancy-related microRNAs in diagnosis or screening.

Reviewer 2 ·

Basic reporting

1) The report is well written, without language mistakes, structure of the report is standard, according to the journal requirements.
2) Background is clear and on the subject, allowing to understand the results and discussion and raw data are shared.

Experimental design

1) The research described is original, performed by the standard and accepted methodology in the field.
2) In the material and methods section I lack the statistical evaluation and I suggest that you describe how were all the p-values obtained (tests etc).

Validity of the findings

1) The research is interesting and fills a knowledge gap in the field.
2) The statistics should be described more in detail.

Additional comments

Well written and interesting report that paves the way for future possible miRNAs use in the diagnosis or prognosis of complications in obstetrics.

Reviewer 3 ·

Basic reporting

no comment

Experimental design

1. The miR array ( V4.0 microRNA Ready-to-use PCR Human Panel I+II TaqMan microarrays) was not found in Qiagen web, please check the information of commercial product was correct.
2. The requirement of include and exclude patients need to clarify.

Validity of the findings

1. The fold change +_ 95% CI of up-regulated or down-regulated microRNAs should be listed in table 2.
2. Were there any references supporting the microRNAs change profile in the study?

·

Basic reporting

Abstract:
Line 25: try 'useful' in place of ideal;
Line 30: CELSPAC: TNG study? – Provide explanation to acronym
Line 35: Results could be more specific, indicate miRNA pattern for deregulation (up or down?) and statistical outcome.
Line 36: we described A specific signature of - insert 'a'; Line 40: with a wide

Introduction:
Line 47: is blood that REMAINS in the placenta…
Lines 49-53: Could use references, e.g. Burns, E., 2014, 10.1891/1058-1243.23.1.41
Authors Mithal et al (referenced on line 54) seem to refer only to the use of cord blood to predict sepsis.
Line 71: Moreover, PREVIOUS research has shown that …
Line 81: follow-up to? – Typo? Try 'which WILL follow up ON 10 000 children…'

Methods – Sample collection and processing:
We question the sample collection – how was the digestion of miRNA avoided, when samples were allowed to sit for 30 minutes. It seems there was almost an hour at room temperature for RNA to degrade. Contrary to this, the Qiagen’s kit used in this study does instruct sample processing as is presented in this manuscript. To better convince the reader, perhaps other sources could be referenced here, such as: Neyro, V., Elie, V., Médard, Y. and Jacqz‐Aigrain, E., 2018. mRNA expression of drug metabolism enzymes and transporter genes at birth using human umbilical cord blood. Fundamental & Clinical Pharmacology.
Line 89: The current study included 24 mothers from whom we collected umbilical …
Line 96: …and aliquoted into tubes as 250 μL samples and …

Methods – miRNA quantification:
Line 103-104: Could add catalog number, search does not bring up the product!
More information about normalization would strengthen this report, was Livak et al. method used: Livak, K.J. and Schmittgen, T.D., 2001. Analysis of relative gene expression data using real-time quantitative PCR and the 2− ΔΔCT method. Methods, 25(4), pp.402-408?
Were spike values taken into account when normalized, as suggested by Chen et al.: Chen, K., Hu, Z., Xia, Z., Zhao, D., Li, W. and Tyler, J.K., 2016. The overlooked fact: fundamental need for spike-in control for virtually all genome-wide analyses. Molecular and Cellular Biology, 36(5), pp.662-667?
Also please add details on statistical analysis following correlation with maternal factors: how was significance calculated? On that end, how was correlation performed?
Line 108: 50% not 50 %

Experimental design

Results:
Reviewer feels it would strengthen manuscript greatly if a scientific basis were delineated for these innate concepts: global profiling > statistical analysis > signature . What do these terms really mean? Pattern? Trend? association? causal? Mechanism? In such a narrow scoped scan of 24 plasma samples (technical replicates?) can we talk about a 'signature'? Please define signature mathematically with probability of it meaning something clinically.

Throughout: make number presentation consistent, e.g. write out numbers one to nine, and present numbers over 10 numerically.
Line 112-113: Did all 24 samples contain each of these 752 miRs? Or was the kit capable of recognizing a total of 752 potential species? Please clarify this important distinction.
Line 117: associated with maternal age NAMELY miR−137… ? Check sense, as 'thereof' is not clear in this context.

Validity of the findings

Discussion:
Line 154: Please clarify following sentence: ‘Despite negative results, this study delineates role of miRNA within delivery course and success.’
I would also suggest making it longer and putting results in the context of known literature, e.g. this would be a good spot to highlight some of the significantly correlated miRNAs and cite relevant functional studies, e.g. the best of the 333 'plasma papers' that cite Weber et al 2010 in Clinical Chemistry.
Line 154: Despite negative results - do you mean previous publications, or your Table 2? If the latter, then specify which entries are positive and which ones are negative. Since no Hypothesis was delineated at the outset of the Experimental Design, then how can there be a "negative result"?
Line 156: "we expected that possible deregulation should be influenced by…" - Finally, this phrase is hinting at an a priori Hypothesis. Trouble, pain, questions, confusion all may lead to a Problem that is normally described in Background Literature Review. After the Problem is addressed then scientific inquiry proceeds to a Rationale and Hypothesis > Objectives >> Specific Aims... Placing the "expected possibilities" at the beginning of the Introduction would help Reader understand the potential usefulness of end-of-pregnancy plasma profiles.

Conclusions:
Line 161: "levels in … plasma are related to … characteristics" To Reader this statement appears to be an indication of POSITIVE results so it flies in the face of Line 154's "negative results". Please clarify.
162: Over interpretation? Have any of the miRNAs that were found deregulated been investigated in relation to maternal health, pathologies or complications?

Overall:
We think this study could offer a good resource for future research. We appreciate the supplementary table with raw reads, which might prove useful to other groups. We might suggest adding to a repository (e.g. GEO) for easier access. Literature search of miRNA and cord blood indicates that not many labs have explored cord blood’s general correlations to maternal health in the context of miRNA. This research would be a good addition to the community if: 1) data is uploaded to a repository; and 2) correlation and statistical analyses are presented in a more transparent and explanatory way.

Additional comments

Micro RNA means shorter than messenger RNA or transfer RNA, but is does not follow that another report on microRNA diversity of blood has to be 'micro' in manuscript length. The fact that the Introduction has just ~390 words implies that there is not much to report, or otherwise the subject matter of the experiment conducted is so self-evident or banal that it is not worth much of an overture. Results reported on Lines 120-134 state that a total of 74 miRs appeared changed in the two dozen samples of cord blood. How do these 74 species compare to miRs detected in other body fluids? Perhaps that question has no 'statistical power' for the approximately 2000 miR genes in the human genome of ~18k other genes? The repertoire of 'array hits' could be on a genomic-scale yet the Introduction does not present the experiment in a Hypothesis-driven approach. A shotgun snapshot strategy makes this random survey look too much like a fishing expedition, an observational 'systems biology' scan of a non-hypothesis small sample of newborn infants.
Weber et al's landmark 2010 report in Clin Chem has amassed over 1500 citations (Scholar), 1032 times cited in Clarivate's Web of Science. Given this plethora of attention one would expect a 2019 report to devote more than one line (Line 59-61) to miR biomarkers, especially given the fact that Weber et al. examined the presence of miRNAs in 12 human body fluids and urine samples from five (n=5) women in different stages of pregnancy. Their Table 3 lists 14 miRNA species that were uniquely detected in plasma so there should be ample opportunity to discuss this present manuscript in light of Weber and subsequent findings. How many hits from Juracek et al's Table 2 occur in the original 2010 report? Of these citers of Weber, 330 deal with plasma, 45 of these in the past twelve months, so there is no paucity of papers with which to compare these present UCB results from Juracek.
Reviewer has heard of an expression, the least publishable unit (LPU), and this manuscript coming in at Methods 365 words, Results 340, Discussion 195 should not be competing for such an accolade. If CELSPAC is in the same league as CALIPER (CIHR: Closing the Gaps in Pediatric Reference Intervals) then each of its publications should be broad, thorough and as deep as possible. Expand and cover the foundational literature on plasma miR profiling in clinical settings. Blood is blood. e.g. But these three smokers' compare how with the international profiles of smoker plasma profiles? Might you try to mine your data more deeply and meaningfully?

---

## Round 0.2 · accepted · Accept

The authors have satisfied the critiques of the reviewers. The current format is ready for publication. Congratulations!

#